# Factors related with the incidence of acute respiratory infections in toddlers in Sleman, Yogyakarta, Indonesia: Evidence from the Sleman Health and Demographic Surveillance System

**Fathmawati Fathmawati**[1]*, **Saidah Rauf**[2], **Braghmandita Widya Indraswari**[3]

**1** Department of Environmental Health, Politeknik Kesehatan Kemenkes Pontianak, Pontianak, Indonesia, **2** Masohi Nursing Study Program, Politeknik Kesehatan Kemenkes Maluku, Ambon, Indonesia, **3** Faculty of Medicine, Public Health and Nursing, Universitas Gadjah Mada Yogyakarta, Yogyakarta, Indonesia

* fathmawati@poltekkes-pontianak.ac.id, fathmawati.ema@gmail.com

**Data Availability Statement:** Sleman HDSS is fully funded and conducted by Faculty of Medicine, Public Health and Nursing (FKKMK), UGM. Thus,

## Abstract

Various factors associated with Acute Respiratory Infections (ARI) in toddlers have been widely observed, but there are no studies using data from the Sleman Health and Demographic Surveillance System (HDSS). This study aimed to determine the factors associated with ARI in children under five in Sleman, Yogyakarta, Indonesia. This research was an observational analytic study with a cross-sectional design, using secondary data from the Sleman HDSS. Data of 463 children under five who met the inclusion and exclusion criteria were used in this study. Inclusion criteria were toddlers who have complete observed variable data. The variables observed were the characteristics of children under five, the attributes of the mother, the physical condition of the house, the use of mosquito coils, sanitation facilities, and sources of drinking water. The exclusion criteria were toddlers with pulmonary tuberculosis in the past year. Data analysis used chi-squared tests for bivariate analysis and multivariate logistic regression analysis. The results showed that working mothers had a greater risk of ARI under five children with OR 1.46 (95% CI = 1.01–2.11), and groundwater as a water source was a protective factor against the occurrence of ARI in toddlers with OR 0.46 (95% CI = 0.26–0.81). After a logistic regression analysis was performed, only the drinking water source variable had a statistically significant relationship with the incidence of ARI in children under five with OR = 0.47 (95% CI = 0.268–0.827). Research on the relationship between water quality and the incidence of ARI in children under five is needed to follow up on these findings.

## Introduction

Acute respiratory infections (ARI) are diseases caused by transmittable agents. Usually, symptoms appear quickly, ranging from a few hours to days after transmission. The spectrum of

Sleman HDSS data use follows the terms and conditions implemented by FKKMK UGM. During the first 2 years after data collection, data are only accessible for Sleman HDSS' researchers and academics community in FKKMK UGM. Afterwards, broader academic communities (e.g., from other universities) are welcome to use Sleman HDSS data. The terms and conditions of Sleman HDSS data use require every user to send the data use application form and brief proposal to Sleman HDSS office for review and to apply for ethical clearance from UGM Medical and Health Research Ethics Committee (MHREC). The applicant will receive the dataset after Sleman HDSS' reviewers and MHREC approve their application. The data set provided by Sleman HDSS does not include any information that can be used to identify the respondents. Furthermore, the data use terms and conditions also prohibit the data user to make any attempts to identify the respondents. The main purpose of the "restriction" in Sleman HDSS data use is to protect Sleman HDSS' respondent personal information. In addition, it also prevents the data use for commercial purposes and prevents duplication in publications based on Sleman HDSS data. Further questions regarding the research ethics of Sleman HDSS data use can be directed to the UGM Medical and Health Research Ethics Committee (mhrec_fmugm@ugm.ac.id; 0274 588688 extension 17225, +62811-2666-869). Detailed information regarding the procedure of data use and application for data use is available in https://hdss.fk.ugm.ac.id/. Further questions can be sent to hdssjogja.fkkmk@ugm.ac.id.

**Funding:** The author(s) received no specific funding for this work.

**Competing interests:** The authors have declared that no competing interests exist.

ARI symptoms can vary, usually including fever and cough. Sore throat, coryza, shortness of breath, wheezing, or difficulty breathing are common presentations. The pathogens related to this disease include pneumococcal and tubercular bacteria as well as influenza and parainfluenza virus, rhinovirus, respiratory syncytial virus (RSV) and severe acute respiratory syndrome coronavirus (SARS-CoV) [1].

ARI is one of the causes of high morbidity and mortality of children under five in the world, including Indonesia [2]. The under-five mortality rate due to ARI in Indonesia ranks first compared to other ASEAN countries. Based on the mortality survey in Indonesia, it shows that ARI is the most common cause of under-five mortality, with a percentage of 22.30% of all under-five deaths. The results of the 2018 Basic Health Research show that the under-five age group is ranked as the highest group of all ARI cases [3]. Another research showed that the highest incidence of ARI was in children aged 0–4 years, ranging from 82 to 114 per 100,000 population [4]. A study conducted one year earlier by the Sleman Health and Demographic Surveillance System (HDSS) Team of the Faculty of Medicine, Nursing and Public Health at the Universitas Gadjah Mada in Yogyakarta, Indonesia also showed the same results [5]. Currently, ARI is still a serious problem in developing counties because it can cause serious morbidity and death in children under five [6, 7].

Some researchers have found several factors that are associated with the incidence of ARI in children under five including nutritional status [8], immunization [8, 9], exclusive breastfeeding, caregivers, exposure to cigarette smoke during pregnancy [10], density occupancy, income, smoking behavior in the family, mother's education [11], use of fuel for cooking [12, 13], and pesticides [14]. Other studies have shown that improving the quality of clean water, sanitation, hygiene, and nutrition can reduce the incidence of ARI in children under five [15]. Humid and moldy house conditions have been related with the incidence of ARI in children hospitalized in New Zealand [16], while lack of lighting and ventilation in the house have been also associated with this incidence of ARI [17].

Research that uses the Sleman HDSS data on factors associated with ARI in children under five has never been conducted. The results of the 2017 Sleman HDSS showed that ARI was ranked first in infectious diseases in Sleman and the highest ARI cases occurred in children under five [5]. If this condition continues, it will cause impaired growth and development in toddlers. This is the first report from Indonesia using the Sleman HDSS data which examined factors associated with the incidence of ARI in under-five children in Yogyakarta, Indonesia. The findings of this study will aid local governments in developing policies for the management of ARI in children under five.

## Methods

### Study design

This research was an analytic observational study with a cross-sectional design. The data analyzed were secondary data collected by the Sleman HDSS team from 2015 (Wave 1) to 2017 (Wave 3) to examine factors related to the incidence of ARI in children under five in the Sleman Regency in Yogyakarta, Indonesia. The Sleman HDSS is the only surveillance system in Sleman that collects data on population transitions, health status and social transitions periodically since 2015. It aims to provide valid data to measure trends, predict baseline demographics, health parameters, and to assess the effects of interventions. The Faculty of Medicine, Public Health, and Nursing UGM collaboration with the Sleman Regency Government conducted this survey. Information from the Sleman HDSS supports evidence-based policymaking [18].

## Population and data collection

A previous study has described the selection of the household as a research unit [18]. The Sleman HDSS research area covers urban, sub-urban and rural areas. The study population was all toddlers aged 13–59 months from HDSS data in 2017. This study used a population of children under five suffering from ARI with inclusion and exclusion criteria. Overall, there were 549 children under five suffering from ARI. Only 463 children under five met the inclusion criteria. The inclusion criteria were: households that had complete family data on socio-demographic conditions, use of mosquito coils, physical conditions of houses and sanitation facilities, as well as immunization status. The exclusion criteria were: households with children who had been diagnosed with pulmonary tuberculosis by health personnel such as doctors, nurses, and midwives in the last year.

Demographic data including characteristics of children under five and maternal characteristics were obtained from HDSS 2017 data. Immunization status and use of mosquito coils were obtained from HDSS 2016 data. Meanwhile, data on house characteristics including type of floor, number of people occupying the house, floor area, house walls, and ceilings, as well as sanitation facilities and water sources were obtained from the 2015 HDSS data.

## Outcome measure

The ARI variable was identified as the child having been diagnosed by health workers with ARI within one month before the survey or based on reports of mothers/respondents on the child complaints of fever accompanied by a cough with phlegm/dry or runny nose within one month before the survey.

## Independent variables

We categorized all variables into two categories, except immunization status. The gender groups of children were male and female. The toddler age groups were <48 months and $\geq$ 48 months. Immunization status consisted of not immunized, incomplete, and complete. Toddlers were classified as completed immunization if they had completed primary vaccinations by the age of 12 months. Low birth weight was classified as yes or no. We used productive age to categorize maternal age. Not attend school—Senior high school ($\leq$ 9 years) was the boundary for the maternal education group, so mothers were grouped into primary education and higher education. Working and not working were the categories of maternal occupation.

We classified types of floors as either soil or cement/ceramic/tile. Grouping of occupancy density per person was based on SNI 03-1733-2004 [19], namely $< 9 \ m^2$/person and $\geq 9 \ m^2$/person. Classification of wall types was as non-cement or cement, and ceilings as no ceiling or had ceiling. The use of mosquito coils was classified as yes or no. Ownership of the latrine and septic tank was either no or yes. Drinking water sources were either groundwater/spring or tap water/refill depot.

## Statistical analysis

Bivariate analysis using the chi-square tests was conducted on independent variables that were suspected to be associated with ARI in children under five. The next step was to conduct a multivariate logistic regression analysis to obtain an appropriate model for the incidence of ARI in children under five with odds ratio (OR) and 95% confidence interval (CI). The significance of the test refers to the $p$-value $< 0.05$.

### Ethical approval

This research was approved by the Medical and Health Research Ethics Committee (MHREC) Faculty of Medicine, Public Health and Nursing Universitas Gadjah Mada-Dr. Sardjito General Hospital (Ref: KE/FK/0987/EC/2020).

## Results

Table 1 shows that there were 217 children under five who had complaints or were diagnosed with ARI. The results of the chi-squared tests on each variable showed no difference in the sex of children under five (OR 1.08; 95% CI = 0.75–1.57), age of toddler (OR 0.99; 95% CI = 0.69–1.46), history of low birth weight (OR 1.02; 95% CI = 0.99–1.04), immunization status (OR 0.96; 95% CI = 0.51–1.82), maternal age (OR 0.91; 95% CI = 0.63–1.31), mother's education (OR 0.96; 95% CI = 0.62–1.49), floor type (OR 0.13; 95% CI = 0.77–1.68), space area per person (OR 1.02; 95% CI = 0.51–1.65), type of wall (OR 0.48; 95% CI = 0.12–1.87), ceiling (OR 0.82; 95% CI = 0.57–1.19), use of mosquito coils (OR 0.75; 95% CI = 0.48–1.18), ownership of latrines (OR 0.89; 95% CI = 0.47–1.67), nor ownership of septic tanks (OR 0.58; 95% CI = 0.30–1.15) between the respondent groups with ARI and those without ARI. There were only significant differences between the groups of respondents with ARI and not ARI in maternal occupation (OR 1.46; 95% CI = 1.01–2.11) and the water source used (OR 0.46; 95% CI = 0.26–0.81).

After the variables of maternal occupation and drinking water sources were subjected to a logistic regression test, only drinking water sources had a $p$-value < 0.05 with an adjusted OR 0.47; 95%CI = 0.268–0.827. The results showed that the variable groundwater/spring water as a source of drinking water is a preventive factor in the occurrence of ARI in children under five after adjusted with the maternal occupation.

## Discussion

### Toddlers' characteristics

Although there was no statistically significant difference, the incidence of ARI in this study was found to be slightly higher in boys than in girls. Researches conducted in Bangladesh and Iraq found that boys had higher ARIs than girls [20–22]. An immunological study showed that women are more immune to pathogens than men, but the severity of infection is higher in women [23]. Another study showed that the incidence of other ARI such as tonsillitis and sinusitis was higher in women in all age groups due to differences in the anatomy of the respiratory tract between men and women [24].

Toddler age < 48 months is a factor in preventing ARI. This study divided toddlers' ages based on readiness to go to playgroups. We hypothesized that toddlers ≥ 48 months of age will be more susceptible to ARI because if they join a playgroup, the risk of transmission is not only from their home environment. However, this age did not show a statistically significant relationship with ARI. This finding is not in line with previous studies [25–27], which included toddlers at a younger age than this study. Interestingly, the odds of ARI in toddlers decreased as the age increased [28]. It was demonstrated that the immaturity of the immune system [29] in younger children (< 24 months) made them more vulnerable to suffer ARI [26, 30–32]. There is no relationship between the age of toddlers and the incidence of ARI in this study because the toddlers involved in this study were aged 25–59 months who have an immune system that is starting to mature [33], so there is no age variation that can distinguish the relationship with ARI.

**Table 1. Socio-demographic, home characteristics and sanitation facilities of the study participant.**

| Variables | ARI (n = 217) | Not ARI (n = 246) | OR (95%CI)[a] | OR (95%CI)[b] |
|---|---|---|---|---|
| | n (%) | n (%) | | |
| Sex of the toddler | | | 1.08 (0.75–1.57) | |
| • Boys | 113 (52) | 123 (50) | | |
| • Girls | 104 (48) | 123 (50) | | |
| Age of the toddler (min. 25 months; max. 59 months) | | | 0.99 (0.69–1.46) | |
| • < 48 months | 135 (62) | 153 (62) | | |
| • ≥ 48 months | 82 (38) | 93 (38) | | |
| Low birth weight | | | 1.02 (0.99–1.04) | |
| • Yes | 17 (7.8) | 16 (6.5) | | |
| • No | 199 (91.7) | 225 (91.5) | | |
| • Missing data | 1 (0.5) | 5 (2) | | |
| Immunization | | | | |
| • No immunization | 79 (36) | 82 (33) | Reference | |
| • Not complete | 113 (52) | 139 (57) | 1.04 (0.78–1.39) | |
| • Complete | 25 (12) | 25 (10) | 0.96 (0.51–1.82) | |
| Age of the mother (min. 21; max. 52) | | | 0.91 (0.63–1.31) | |
| • ≤ 35 years | 112 (52) | 133 (54) | | |
| • > 35 years | 105 (48) | 113 (46) | | |
| Mother's education | | | 0.96 (0.62–1.49) | |
| • Basic education | 48 (22) | 56 (23) | | |
| • Higher education | 169 (78) | 190 (77) | | |
| Mother's occupation | | | 1.46* (1.01–2.11) | 1.44 (0.99–2.08) |
| • Not working | 114 (53) | 106 (43) | | |
| • Working | 103 (47) | 140 (57) | | |
| Floor type | | | 0.13 (0.77–1.68) | |
| • Soil | 2 (1) | 0 (0) | | |
| • Ceramic/tile/cement | 215 (99) | 246 (100) | | |
| Occupancy density | | | 1.02 (0.51–1.65) | |
| • < 9 m²/person | 18 (8) | 20 (8) | | |
| • ≥ 9 m²/person | 199 (92) | 226 (92) | | |
| House wall | | | 0.48 (0.12–1.87) | |
| • Non cement | 3 (1) | 7 (3) | | |
| • Cement/brick | 214 (99) | 239 (97) | | |
| Ceilings | | | 0.82 (0.57–1.19) | |
| • No ceiling | 105 (48) | 131 (53) | | |
| • Have ceiling | 112 (52) | 115 (47) | | |
| Using mosquito coil | | | 0.75 (0.48–1.18) | |
| • Yes | 40 (18) | 57 (23) | | |
| • No | 177 (82) | 189 (77) | | |
| Have a latrine | | | 0.89 (0.47–1.67) | |
| • No | 19 (9) | 24 (10) | | |
| • Yes | 198 (91) | 222 (90) | | |
| Have a septic tank | | | 0.58 (0.30–1.15) | |
| • No | 14 (6) | 26 (11) | | |
| • Yes | 203 (94) | 220 (89) | | |
| Water source | | | 0.46* (0.26–0.81) | 0.47* (0.27–0.83) |
| • Ground water/spring | 179 (82) | 224 (91) | | |

(*Continued*)

**Table 1.** (Continued)

| Variables | ARI (n = 217) | Not ARI (n = 246) | OR (95%CI)[a] | OR (95%CI)[b] |
|---|---|---|---|---|
| | n (%) | n (%) | | |
| • Tap water/refill water | 38 (18) | 22 (9) | | |

ARI, acute respiratory infection; CI, confidence interval; OR, odds ratio;

[a] = Unadjusted OR;

[b] = Adjusted OR, adjusted for mother's occupation and water source;

* = significant at $p < 0.05$.

This study did not find a significant relationship between low birth weight and ARI. A longitudinal study in Bangladesh also found no significant association between birth weight and ARI [22]. However, low birth weight correlates with the incidence of ARI [34]. Some researchers have shown that infants with low birth weight have organs that are not fully developed and are very susceptible to infection [35, 36]. There was no relationship between birth weight and ARI in this study due to the few cases of low birth weight found. The Sleman Regency Government seeks to reduce the low birth weight rate by minimizing the anemia rate of pregnant women for the health of mothers and babies [37].

This study found that only 11% of children under five received complete immunization. This result is different from the 2018 Basic Health Research, which showed that Yogyakarta Province is ranked second after Bali in terms of universal immunization coverage that exceeds the national coverage rate [3]. This study did not show a significant relationship between immunization status and ARI. This is in contrast to studies conducted in two other regions in Indonesia which showed a significant relationship between immunization status and ARI [8, 9]. The absence of a relationship between immunization status and ARI may be due to the low immunization coverage in this study. The issue of vaccine rejection in Sleman is undeniable [37, 38]. Local governments should prioritize increasing public understanding of vaccines so that immunization coverage can be achieved.

## Mothers' characteristics

This study found that there was no significant association between a mother's education level with the incidence of ARI. This result was inconsistent with previous research [13, 22], which reported that higher maternal education might promote household hygiene, thus leading to the decline of developing ARI in children. The discrepancy in results could be attributed to the differences in the mother's education categorization. The odds of ARI in children under five in Ethiopia and Turkey whose mothers had no formal education were increased compared with children whose mothers completed secondary school [26]. However, in addition to not subgrouping maternal education, our study did not classify mothers who did not attend formal schools. The education of the people of Sleman Regency is high. Sleman has a Human Development Index (HDI) higher than the national average. One of the components is the average length of schooling for residents aged 25 years and over which reaches 9.32 years. This figure is higher than the national figure of 8.17 years [39].

The unadjusted OR value of maternal occupation showed that children with working mothers have a 1.46 times higher risk of developing ARI. After adjusting for the source of drinking water, maternal occupation no longer exhibits a significant relationship, but it remains a contributing factor. Working mothers generally have higher education. A study found that women working in Sleman tend to entrust their children to daycare or leave them with family/caregivers while working [40]. The risk of transmitting ARI is greater because toddlers will

interact with each other during this time away from home. Another study conducted in Sleman showed that working mothers were less likely to give exclusive breastfeeding [41]. One of the reasons is the lack of workplace support to provide space for nursing mothers [42]. Even though the Sleman Regency Government has issued a regulation requiring workplaces to provide lactation rooms for breastfeeding mothers, some companies have not fully complied with it. This is because business owners do not understand why this regulation was issued or have not received socialization regarding this matter [43]. This study did not have data on exclusive breastfeeding that could support these findings. This finding is in line with previous research which found working mothers have a positive correlation with the incidence of ARI [44].

## Physical house condition

The floor type of the respondents' house showed no correlation to ARI in children under five in Sleman. Some studies had different results. They found a significant relationship between floor type and the incidence of ARI in children under five [45, 46]. Research conducted in Cirebon, Indonesia found a weak relationship between floor types and the incidence of ARI [47]. In this research, almost all floors of the respondents' houses are ceramic/tile/cement and watertight. We found only two toddlers with ARI whose houses have soil floors, and this variable was not related to the incidence of ARI. Even so, the floors made of the earth tend to become dry and dusty more easily and difficult to clean, which can interfere with breathing.

This study found that there was no association between occupancy density with the incidence of ARI. This result is different from the several studies [11, 45, 46, 48] that found a relationship between house density and the incidence of ARI in children under five. House conditions with an occupancy density of $<9$ m$^2$/person only occurred in less than 9% of the observed houses. Most of the respondents' dwellings have a sufficient area. Residential density measurements in this study are based on an extensive house divided by the number of residents, not including spacious bedrooms and the number of people who occupy them. This point is important considering that ARI is easily transmitted in densely populated situations [49]. A study conducted in Padang, Indonesia found that residential areas with high density were almost 22 times more likely to have ARI in children under five [50].

The type of wall was not related to the incidence of ARI in children under five in Sleman. This result is consistent with research conducted in Surabaya [17]. Types of walls made of wood or woven bamboo are difficult to clean because of their rough surface texture. In this study, only 2% of houses had non-cement wall material. Almost all the dwellings of the respondents already have walls with a solid material that is easier to clean. This condition explains that there is no relationship between the walls of the house and ARI.

The study did not find a relationship between the type of house ceiling and the incidence of ARI. Another study has also found the same result [51]. The ceiling functions to keep dust from entering the house through the gaps between the walls and the roof. This research found that the proportion of houses that have ceilings and do not have a ceiling was almost equal. Likewise, the proportion of children under five with ARI and not with ARI was almost evenly distributed in the groups of houses with and without ceilings. Both groups have the same opportunity for ARI. The ceiling functions to keep dust from entering the house through the gaps between the walls and the roof. Unfortunately, this study did not have data on indoor air quality. ARI in toddlers in Sleman could be caused by polluted air quality. The results of one research showed that the Sleman urban area is an area with the highest air pollution compared to other urban areas in Yogyakarta Province. The carbon monoxide (CO) content in the city of Sleman, namely Depok, Ngaglik and Kalasan, exceeds the quality standard by up to seven times the safe limit [52]. Several studies found that green open spaces in Sleman are not yet in

line with public safety and city planning needs [53–55]. If adequate green open spaces are available, they can absorb pollutants from the air.

## The use of mosquito coil

The use of mosquito coils in this study did not show a significant relationship. This result is different from previous studies that found a relationship between the use of mosquito coils and ARI [56, 57]. The difference between this study and previous studies is the proportion of mosquito coil users in the research subjects. In this study, mosquito coils were not popularly used by respondents, only about 20%. Meanwhile in previous studies, more than 50% of respondents used mosquito coils.

## Water and sanitation facilities

Sanitation facilities in the form of toilet and septic tank ownership were not related to the incidence of ARI in children under five in Sleman. One recent study found that improved sanitation can reduce the incidence of ARI [15]. Another study found a relationship between the use of septic tanks with the incidence of ARI [20]. More than 90% of the respondents in this study already have a latrine and a septic tank. This result is in line with the results of the 2018 Basic Health Research which showed that more than 95% of households in Yogyakarta Province have disposed of their feces properly [3]. Sanitation facilities are not a factor in the occurrence of ARI in children under five in Sleman.

This study found that ground water or spring water as the source of water used by respondents for eating and drinking was a preventive factor for ARI. Sleman has a high potential for groundwater and springs, especially in rural areas [58, 59]. The Sleman Regency Government divides its territory into four sectors. Two sectors namely the Merapi Slope Area and the West Region, have abundant water resources [60], and groundwater quality is relatively better than other areas in Yogyakarta Province with low nitrate content [61, 62]. Improving water quality can reduce the incidence of ARI in children under five according to studies conducted in Rwanda [63] and Bangladesh [15].

The service coverage of the Local Water Company (LWC) in Sleman only reaches 23% of the population [64], and while a small portion uses commercially managed refill drinking water, the rest use groundwater or spring water as the main source of drinking water. The results of the 2015 Water Quality Survey in the Special Region of Yogyakarta stated that more than 60% of the people of Sleman use groundwater as a source of drinking water [65]. Therefore, in this study, it was found that there is only a small proportion of respondents whose source of drinking water was not groundwater/spring water.

The quality of water produced by LWC in Sleman has not met the requirements set by the Ministry of Health. Several parameters (including *Coliform* and *E. coli*) were found to exceed the required levels [64, 66, 67]. Likewise, the quality of refill drinking water in the Sleman area has not met the requirements. Earlier research found high content of *Coliform* and *E. coli* in samples examined in both raw water and processed water in refill drinking water depots [68, 69]. The Water Quality Survey in Yogyakarta Province in 2015 showed that only 8.5% of households could access safe drinking water according to the Sustainable Development Goals (SDGs) [65].

The presence of *E. coli* in drinking water can trigger lung damage in mice [70], but further exploration is still needed to explain this pathogenesis [71]. Meanwhile, research conducted in Rajasthan, India showed that consuming drinking water that contains high nitrates is significantly associated with the incidence of recurrent ARI. This study explained that consuming water containing nitrates will increase the production of methemoglobin and free radicals of

nitric oxide and oxygen that will lead to alveolar damage and improper ventilation and perfusion, which may be the cause of death in children due to recurrent ARI [72]. Regrettably, research conducted recently did not include nitrates in the examined parameters [15, 63]. Accordingly, further research is needed to be able to explain the relationship between improving water quality and reducing the incidence of ARI.

## Research limitations

The incidence of ARI in toddlers in this study was only based on the mother's memory, which is prone to recall bias. Additionally, the Sleman HDSS has no data on ventilation, which functions to change the air in the house so that residents get fresh air. Increasing air circulation in the dwelling through ventilation and windows is a factor that can reduce the incidence of ARI [73]. Natural lighting in homes and bedrooms is related to temperature and humidity. Both of these are influential factors in the proliferation of the agents causing ARI [16]. Data on smokers inside the home are critical to include in the analysis because it is closely related to indoor air quality, likewise the data are limited on the use of firewood for cooking, which should be included considering some people still use it.

## Conclusions

Respondents who use tap water or refilled drinking water depots are at risk of experiencing ARI. The following Sleman HDSS survey must pay attention to the quality of water consumed by respondents, especially *E. coli* and nitrates. It is to ensure the relationship between the use of water sources and the incidence of ARI in children under five so that the government can determine the right policy in reducing the morbidity of ARI in children under five.

## Acknowledgments

The authors are grateful to Sleman HDSS Management for allowing us to use Sleman HDSS data; and Klinik Bahasa Faculty of Medicine, Public Health and Nursing UGM for proofreading this manuscript.

## Author Contributions

**Conceptualization:** Fathmawati Fathmawati, Saidah Rauf, Braghmandita Widya Indraswari.

**Data curation:** Fathmawati Fathmawati, Saidah Rauf.

**Formal analysis:** Fathmawati Fathmawati, Braghmandita Widya Indraswari.

**Methodology:** Fathmawati Fathmawati.

**Writing – original draft:** Fathmawati Fathmawati.

**Writing – review & editing:** Fathmawati Fathmawati, Saidah Rauf, Braghmandita Widya Indraswari.

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
