## [Decision Letter · Decision Letter 0]

21 Apr 2021

PONE-D-20-38919

Factors related with the incidence of acute respiratory infections in toddlers in Sleman, Yogyakarta, Indonesia: Evidence from The Sleman Health and Demographic Surveillance System

PLOS ONE

Dear Dr. Fathmawati,

Thank you for submitting your manuscript to PLOS ONE. After careful consideration, we feel that it has merit but does not fully meet PLOS ONE’s publication criteria as it currently stands. Therefore, we invite you to submit a revised version of the manuscript that addresses the points raised during the review process.

We look forward to receiving your revised manuscript.

Kind regards,

Muhammad Aziz Rahman, MBBS, MPH, CertGTC, GCHECTL, PhD

Academic Editor

PLOS ONE

Journal Requirements:

3a) If there are ethical or legal restrictions on sharing a de-identified data set, please explain them in detail (e.g., data contain potentially identifying or sensitive patient information) and who has imposed them (e.g., an ethics committee). Please also provide contact information for a data access committee, ethics committee, or other institutional body to which data requests may be sent.

3b) If there are no restrictions, please upload the minimal anonymized data set necessary to replicate your study findings as either Supporting Information files or to a stable, public repository and provide us with the relevant URLs, DOIs, or accession numbers. Please see http://www.bmj.com/content/340/bmj.c181.long for guidelines on how to de-identify and prepare clinical data for publication. For a list of acceptable repositories, please see http://journals.plos.org/plosone/s/data-availability#loc-recommended-repositories.

This paper uses data from Sleman HDSS (Health and Demographic Surveillance System)

wave 1, 2, 3. The Sleman HDSS data collection has been primarily funded by Faculty of

Medicine, Public Health, and Nursing, Universitas Gadjah Mada, Yogyakarta, Indonesia.

Additional Editor Comments:

Please address the comments of both reviewers for our further consideration.

Reviewers' comments:

Reviewer's Responses to Questions

**Comments to the Author**

1. Is the manuscript technically sound, and do the data support the conclusions?

Reviewer #1: Partly

Reviewer #2: Yes

2. Has the statistical analysis been performed appropriately and rigorously? 

Reviewer #1: No

Reviewer #2: Yes

3. Have the authors made all data underlying the findings in their manuscript fully available?

Reviewer #1: Yes

Reviewer #2: Yes

4. Is the manuscript presented in an intelligible fashion and written in standard English?

Reviewer #1: Yes

Reviewer #2: Yes

5. Review Comments to the Author

Reviewer #1: 1. In the introduction, researchers clearly mentioned the problem statement but did not justify why it is important to conduct this study in a HDSS site; they also did not mention how the HDSS-based study findings will contribute in policy making.

2. The experiment was not conducted rigorously. The authors did not mention how they calculated the sample size and why they used the purposive sampling methods in a cross-sectional observational study. Purposive sampling is suitable for qualitative research.

3. In the study design, they mentioned that they analyzed the secondary data from 2015 to 2017 but at the line 63, they mentioned that the study population was from HDSS data in 2017. They need to explain this discrepancy clearly. They also did not mention whether this HDSS site is urban or rural based.

4. Except the source of drinking water as a protective factor, the study did not find the significant relationship with any other important predictors like mothers’ education level and occupation, occupancy density, physical condition of house, use of mosquito coil. This study finding is also inconsistent with many other study findings which is not explained properly by the authors. This contradiction may be due to use of inappropriate sampling method. Though authors decided to select the appropriate statistical analysis method but it is not appropriate to run the statistical analysis using a purposive sample in a cross-sectional observational study.

5. Authors mentioned that data can’t be shared without the permission of the management team but data which are available in the table is sufficient to understand the study findings

6. In the conclusion, they did not make any specific recommendation to the policy makers.

7. Nonetheless the manuscript is presented in an understandable way, still it needs to improve further. Such as:

a. authors mainly compared this study finding with African-based studies instead of Asian countries; it may affect the policy makers to make the applicable decision

b. Typo: prevalence at the line 128, it should be incidence

c. Line 152, they mentioned “similar results were reported by [8,9,35] and in Line 201, they mentioned “this research is consistent with [17]; this is not the appropriate way to mention it when they will compare their study findings with other study findings.

Reviewer #2: This article need to give more information about the reason why some of independent variables was not have signicantly related to ARI. Please write argumentation for each factor. This contains is very important to ellaborate the condition that different with the other study

6. PLOS authors have the option to publish the peer review history of their article (what does this mean?). If published, this will include your full peer review and any attached files.

Reviewer #1: No

Reviewer #2: No

---

## [Author Response · Author response to Decision Letter 0]

10 Jul 2021

Reviewer#1:

1. In the introduction, researchers clearly mentioned the problem statement but did not justify why it is important to conduct this study in a HDSS site; they also did not mention how the HDSS-based study findings will contribute in policy making.

Response: Thank you for pointing this out. We agree with this comment. Therefore, we have added a description of the importance of this research and its benefits for policymakers. This change can be seen in lines 77 - 83.

2. The experiment was not conducted rigorously. The authors did not mention how they calculated the sample size and why they used the purposive sampling methods in a cross-sectional observational study. Purposive sampling is suitable for qualitative research.

Response: Thank you for pointing this out. We used the entire population of children under five in the Sleman HDSS data that matched the inclusion criteria. We did not include children with exclusion criteria in this study. We have changed the description of the research method. This change is in lines 96 - 99.

3. In the study design, they mentioned that they analyzed the secondary data from 2015 to 2017 but at the line 63, they mentioned that the study population was from HDSS data in 2017. They need to explain this discrepancy clearly. They also did not mention whether this HDSS site is urban or rural based.

Response: Agree. We have accordingly revised the method section by adding a description of the research area and the data obtained. The changes can be seen in lines 96, 105 - 110.

Response: Thank you. For some of the variables, we have added descriptions. However, in terms of occupancy density, we have explained that there is no relationship between occupancy density and the incidence of ARI because only a small proportion of respondents' houses do not meet the standards. Same is the case with the type of wall of the house. We have also revised the methods section on sampling. This study did not take a sample but used data from the entire population of children under five who met the criteria. 

We added some explanations of the cause of the absence of a relationship between several variables studied with ARI in children under five. The changes can be seen in

Lines 173 – 175 (for toddler’s sex).

Line 184 – 187 (for toddler’s age)

Line 188 – 195 (for low birth weight)

Line 199 – 205 (for immunization)

We added some explanations about mother’s education that has no relationship to ARI. The changes can be seen in lines 215 – 219.

We added some explanations about mother’s occupation that has no relationship to ARI. The changes can be seen in lines 229 – 235.

We added some explanations about occupancy density in line 254 – 255.

We added some explanations about ceilings in line 267 – 270.

We revised the explanation about mosquito coils in line 278 – 283.

5. Authors mentioned that data can’t be shared without the permission of the management team but data which are available in the table is sufficient to understand the study findings.

Response: Thank You

6. In the conclusion, they did not make any specific recommendation to the policy makers.

Response: We agree with this and revised the conclusion. It can be seen in line 338 – 348.

7. Nonetheless the manuscript is presented in an understandable way, still it needs to improve further. Such as:

a. authors mainly compared this study finding with African-based studies instead of Asian countries; it may affect the policy makers to make the applicable decision.

Response: We agree with this comment. Therefore, we have replaced and added some research according to your suggestions. 

b. Typo: prevalence at the line 128, it should be incidence

Response: Thank you. We have fixed it.

c. Line 152, they mentioned “similar results were reported by [8,9,35] and in Line 201, they mentioned “this research is consistent with [17]; this is not the appropriate way to mention it when they will compare their study findings with other study findings.

Response: Thank you. We have fixed it.

Reviewer #2: 

This article needs to give more information about the reason why some of independent variables was not have significantly related to ARI. Please write argumentation for each factor. This contains is very important to elaborate the condition that different with the other study.

Response: Thank you. We have incorporated your suggestion throughout the manuscript.

---

## [Decision Letter · Decision Letter 1]

26 Jul 2021

PONE-D-20-38919R1

Factors related with the incidence of acute respiratory infections in toddlers in Sleman, Yogyakarta, Indonesia: Evidence from The Sleman Health and Demographic Surveillance System

PLOS ONE

Dear Dr. Fathmawati,

Thank you for submitting your manuscript to PLOS ONE. After careful consideration, we feel that it has merit but does not fully meet PLOS ONE’s publication criteria as it currently stands. Therefore, we invite you to submit a revised version of the manuscript that addresses the points raised during the review process.

Please address the issue of justification and revise the conclusion section, as advised by the reviewer.

We look forward to receiving your revised manuscript.

Kind regards,

Associate Professor Dr Muhammad Aziz Rahman

Academic Editor

PLOS ONE

Journal Requirements:

Reviewers' comments:

Reviewer's Responses to Questions

**Comments to the Author**

1. If the authors have adequately addressed your comments raised in a previous round of review and you feel that this manuscript is now acceptable for publication, you may indicate that here to bypass the “Comments to the Author” section, enter your conflict of interest statement in the “Confidential to Editor” section, and submit your "Accept" recommendation.

Reviewer #1: All comments have been addressed

2. Is the manuscript technically sound, and do the data support the conclusions?

Reviewer #1: Partly

3. Has the statistical analysis been performed appropriately and rigorously? 

Reviewer #1: Yes

4. Have the authors made all data underlying the findings in their manuscript fully available?

Reviewer #1: Yes

5. Is the manuscript presented in an intelligible fashion and written in standard English?

Reviewer #1: Yes

6. Review Comments to the Author

Reviewer #1: Though, authors tried their level best to answer our queries, still there is room for improvement

a. there is lack of strong justification "why HDSS data is important for local policy makers"; does it mean that there is no other local data except HDSS to help the policy makers to make their decisions?

b. Very weak conclusion; 1st recommendation is related to immunization where authors mentioned that it is not related to ARI; 2nd recommendation is related to breastfeeding which is not supported by study findings/data

7. PLOS authors have the option to publish the peer review history of their article (what does this mean?). If published, this will include your full peer review and any attached files.

Reviewer #1: No

---

## [Author Response · Author response to Decision Letter 1]

6 Sep 2021

Dr. Fathmawati Fathmawati

Department of Environmental Health, 

Politeknik Kesehatan Kemenkes Pontianak, 

Indonesia

Email: fathmawati@poltekkes-pontianak.ac.id or fathmawati.ema@gmail.com

Pontianak, 19 August 2021

Dear Dr. Rahman

Academic Editor PLOS ONE

We have a desire to thank you and the reviewers who allowed us the opportunity to revise our paper entitled "Factors associated with the incidence of ARI in Toddlers in Sleman, Yogyakarta, Indonesia: Evidence from The Sleman Health and Demographic Surveillance System" and provide valuable comments. We think our paper will be better than the previous version after revising according to your precious views and comments. We hope that this manuscript afterward's careful revision meets your high standards. The authors welcome further constructive comments if any.

Below we provide the point-by-point responses. All modifications in the manuscript have been highlighted in yellow.

 

RESPONSES TO ACADEMIC EDITOR

Here are responses to the improvements we had to make:

Journal Requirements:

Response: Thank you for your kind reminder. We have reviewed the references and corrected the retracted papers. To ensure our references are complete and correct, we complete all articles published in the journal with DOI while other references with URLs. In addition, we also added the title of the paper in the original version using Indonesian. We present the details of the changes in the bibliography:

No. Number in 2nd version Number in 3rd version Changes 

1. 1 1 Available: https://www.who.int/publications/i/item/infection-prevention-and-control-of-epidemic-and-pandemic-prone-acute-respiratory-infections-in-health-care

2. 2 2 Available: https://apps.who.int/iris/handle/10665/324835

3. 3 3 Available: https://www.litbang.kemkes.go.id/laporan-riset-kesehatan-dasar-riskesdas/

4. 8 8 doi:10.33846/hn.v1i3.59

5. 9 9 [Cakupan imunisasi dasar dengan kejadian ISPA pada balita usia 1-3 tahun di wilayah Puskesmas Wonosari 1 Kabupaten Gunung Kidul]

6. 11 11 [Faktor-faktor yang mempengaruhi kejadian infeksi saluran pernapasan akut (ISPA) pada anak usia 12-59 bulan di Puskesmas Tebet Barat, Kecamatan Tebet, Jakarta Selatan]

7. 14 14 Tarmure S, Alexescu TG, Orasan O, Negrean V, Sitar-Taut AV, Coste SC, et al. Influence of pesticides on respiratory pathology – A literature review. Ann Agric Env Med. 2020;27: 194–200. doi:10.26444/aaem/121899

8. 27 27 doi:10.4103/0974-777X.107167

9. 37 37 Available: https://dinkes.slemankab.go.id/download

10. 38 38 doi:10.37638/jsk.25.3.1-13

11. 39 39 Available: https://yogyakarta.bps.go.id/pressrelease/2019/05/06/951/indeks-pembangunan-manusia-d-i--yogyakarta-2018.html

12. 40 40 Puspitasari E. The dual role of women in working mothers in Pakembinangun, Pakem, Sleman, Yogyakarta [Peran ganda perempuan pada ibu bekerja di desa Pakembinangun, Pakem, Sleman, Yogyakarta]. Universitas Negeri Yogyakarta. 2016. Available: http://eprints.uny.ac.id/37957/

13. 42 42 Available: http://digilib.unisayogya.ac.id/338/

14. 43 43 doi:http://dx.doi.org/10.21927/ijnd.2019.7(3).89-96

15. 45 Remove 

16. 53 52 Available: http://eprints.ums.ac.id/86266/

17. 56 55 [Hubungan paparan asap dengan kejadian infeksi saluran pernapasan akut (ISPA) pada anak usia 0-5 tahun di wilayah pertanian Kecamatan Panti, Kabupaten Jember]

doi:10.14710/jekk.v5i2.7152

18. 59 58 [Evaluasi potensi mataair untuk kebutuhan air domestik di Kecamatan Cangkringan Kabupaten Sleman pasca erupsi Merapi 2010]

19. 60 59 Available: http://www.slemankab.go.id/profil-kabupaten-sleman/geografi/karakteristik-wilayah

20. 64 63 [Evaluasi kinerja PDAM Sleman di bidang operasi dan pelayanan pada ibukota Kecamatan Prambanan, Kalasan, Ngemplak, Bimomartani, Condong Catur]

Available: https://dspace.uii.ac.id/handle/123456789/16317

21. 65 64 [Hasil Survei Kualitas Air di Daerah Istimewa Yogyakarta 2015]

Available: https://www.bps.go.id/publication/2016/11/03/26dd424f1c7391a6c62adf33/hasil-survei-kualitas-air-di-daerah-istimewa-yogyakarta-tahun-2015.html

22. 68 67 [Evaluasi pengendalian kinerja kualitas air minum pada depot air minum isi ulang di Kabupaten Sleman, Yogyakarta]

 

RESPONSES TO REVIEWERS

Reviewer's Responses to Questions

Comments to the Author

Reviewer #1: Though, authors tried their level best to answer our queries, still there is room for improvement

a. there is lack of strong justification "why HDSS data is important for local policy makers"; does it mean that there is no other local data except HDSS to help the policy makers to make their decisions?

Response: Thank you. We have added the explanation that can be seen in line 88 – 90.

The Sleman HDSS is the only surveillance system in Sleman that collects data on population transitions, health status and social transitions periodically since 2015.

b. Very weak conclusion; 1st recommendation is related to immunization where authors mentioned that it is not related to ARI; 2nd recommendation is related to breastfeeding which is not supported by study findings/data

Response: Thank you for pointing this out. We have changed the conclusion. The changes can be seen in line 339 – 344.

Respondents who use tap water or refilled drinking water depots are at risk of experiencing ARI. The following Sleman HDSS survey must pay attention to the quality of water consumed by respondents, especially E. coli and nitrates. Further research needs to analyze the relationship between water quality and the incidence of ARI. It is to ensure the relationship between the use of water sources and the incidence of ARI in children under five so that the government can determine the right policy in reducing the morbidity of ARI in children under five.

We look forward to hearing from you in due time regarding our submission and to respond to any further questions and comments you may have.

Thanks for your kindly.

Best Regards,

Dr. Fathmawati Fathmawati

---

## [Editor Report · Decision Letter 2]

14 Sep 2021

Factors related with the incidence of acute respiratory infections in toddlers in Sleman, Yogyakarta, Indonesia: Evidence from The Sleman Health and Demographic Surveillance System

PONE-D-20-38919R2

Dear Dr. Fathmawati,

We’re pleased to inform you that your manuscript has been judged scientifically suitable for publication and will be formally accepted for publication once it meets all outstanding technical requirements.

Kind regards,

Associate Professor Dr Muhammad Aziz Rahman

Academic Editor

PLOS ONE

---

## [Editor Report · Acceptance letter]

17 Sep 2021

PONE-D-20-38919R2 

Factors related with the incidence of acute respiratory infections in toddlers in Sleman, Yogyakarta, Indonesia: Evidence from The Sleman Health and Demographic Surveillance System 

Dear Dr. Fathmawati:

I'm pleased to inform you that your manuscript has been deemed suitable for publication in PLOS ONE. Congratulations! Your manuscript is now with our production department. 

Kind regards, 

on behalf of

Associate Professor Dr. Muhammad Aziz Rahman 

Academic Editor

PLOS ONE